# Low-Altitude Infrared Slow-Moving Small Target Detection via Spatial-Temporal Features Measure

**DOI:** 10.3390/s22145136

**Published:** 2022-07-08

**Authors:** Jing Mu, Junmin Rao, Ruimin Chen, Fanming Li

**Affiliations:** 1Key Laboratory of Infrared System Detection and Imaging Technology, Chinese Academy of Sciences, Shanghai 200083, China; mujing@mail.sitp.ac.cn (J.M.); raojunmin@mail.sitp.ac.cn (J.R.); chenruimin@mail.sitp.ac.cn (R.C.); 2Shanghai Institute of Technical Physics, Chinese Academy of Sciences, Shanghai 200083, China; 3University of Chinese Academy of Sciences, Beijing 100049, China

**Keywords:** infrared image sequences, low-altitude slow-moving small target, spatial-temporal features, trajectory continuity

## Abstract

Robust detection of infrared slow-moving small targets is crucial in infrared search and tracking (IRST) applications such as infrared guidance and low-altitude security; however, existing methods easily cause missed detection and false alarms when detecting infrared small targets in complex low-altitude scenes. In this article, a new low-altitude slow-moving small target detection algorithm based on spatial-temporal features measure (STFM) is proposed. First, we construct a circular kernel to calculate the local grayscale difference (LGD) in a single image, which is essential to suppress low-frequency background and irregular edges in the spatial domain. Then, a short-term energy aggregation (SEA) mechanism with the accumulation of the moving target energy in multiple successive frames is proposed to enhance the dim target. Next, the spatial-temporal saliency map (STSM) is obtained by integrating the two above operations, and the candidate targets are segmented using an adaptive threshold mechanism from STSM. Finally, a long-term trajectory continuity (LTC) measurement is designed to confirm the real target and further eliminate false alarms. The SEA and LTC modules exploit the local inconsistency and the trajectory continuity of the moving small target in the temporal domain, respectively. Experimental results on six infrared image sequences containing different low-altitude scenes demonstrate the effectiveness of the proposed method, which performs better than the existing state-of-the-art methods.

## 1. Introduction

With the popularity and widespread use of unmanned aerial vehicles (UAVs) in recent years, the effective surveillance of low-altitude slow-moving small targets represented by UAVs has been a critical problem that needs to be addressed in airspace security [1,2]. Infrared search and tracking (IRST) systems have several merits, such as long-distance detection, day-and-night monitoring, and better aerosols penetration capability [3]. So, they are less affected by environmental illumination and weather conditions, becoming one of the most practical means for low-altitude security [4,5]; however, due to the long imaging distance and low resolution of the infrared sensor, the small target in an infrared image only occupies a few pixels (about 0.12% of the total number of pixels in the image) [6], even one pixel in extreme cases, without a specific shape and detailed texture. Moreover, the low-altitude background has more intricate background interference with high brightness, such as vegetation, buildings, and lanes that easily submerges real targets in the background and results in a low signal-to-noise ratio for infrared images. To keep the observed drones in the field of view (FOV), the infrared detector would shift following the moving targets, which causes the background to change slowly; therefore, effective infrared moving small target detection under complex low-altitude conditions is still a challenging task.

Up to now, researchers have proposed numerous infrared small target detection methods. These methods mainly include two categories: single-frame detection methods and sequential detection methods. The single-frame detection methods only exploit the features of the small target and background in the spatial domain. The small target is often modeled using a 2D Gaussian function because of its isotropic gray distribution, and many local contrast measures using a nine-cell square kernel are proposed to enhance the small target [7,8,9,10]. Moreover, the background properties, such as local consistent and non-local self-correlation, are widely utilized to estimate the background [11,12,13]. Then, the target can be extracted from the difference map between the original image and the estimated background image. The single-frame detection methods require less memory and are easy to implement; however, the available information from a single frame is insufficient to achieve stable infrared small target detection, especially when detecting the point target from cluttered backgrounds. Temporal cues contained in an image sequence, such as the motion consistency of the moving target and the high correlation of the background in adjacent frames, are essential for robust small target detection. Most existing sequential detection methods [14,15,16] combine the local contrast in a single frame and the interframe difference to detect the moving target. The target enhancement ability of these methods mainly relies on the local contrast measure in the spatial domain. In addition, they just exploit local information contained in a few successive frames; therefore, they easily ignore dim targets and are sensitive to random noises. Additionally, these methods would generate numerous false alarms in the case that the background changes with the moving target. In conclusion, existing methods cannot obtain satisfactory performance when detecting the low-altitude slow-moving small target.

To adequately excavate the spatial-temporal information and motion continuity of the moving target, we propose a new sequential detection method based on spatial-temporal features measure (STFM). The main idea of STFM is to enhance the detection ability by integrating the local spatial-temporal features of the small target and improve the detection precision using a multi-frame confirmation. Since the small target can be regarded as a compact area of isotropic distribution in the spatial domain, we first calculate the local grayscale difference (LGD) using a circular kernel and the spatial saliency map is obtained. Then, interframe registration is applied before exploiting the temporal features of the moving target. By analyzing the image sequence after registration, we find that the appearance of the moving target will cause a local variation in the temporal dimension. Thus, a short-term energy aggregation (SEA) mechanism is proposed to obtain the temporal saliency map. The SEA module can accumulate the energy of the small target from a short-term sub-sequence consisting of multiple successive frames, which is essential to enhance the dim moving target. Next, a normalized fusion mechanism is adopted to integrate the results of the LGD and SEA modules, and the spatial-temporal saliency map (STSM) is obtained. Moreover, an adaptive threshold mechanism is utilized to obtain the candidate targets from the STSM. Finally, a long-term trajectory continuity (LTC) measurement is designed based on the fact that the position of the slow-moving target will not abruptly change in a successive image sequence. In summary, the proposed method, consisting of LGD, SEA, and LTC modules, fully utilizes the spatial and temporal features to detect the low-altitude slow-moving small target from infrared image sequences. Experimental results on six real infrared image sequences containing various low-altitude scenes demonstrate that the proposed method outperforms existing state-of-the-art methods both in detection ability and precision. The main contributions of our method can be summarized as follows:(1)A local grayscale difference (LGD) measure based on a circular kernel is proposed to exploit the spatial feature of the small target;(2)A short-term energy aggregation (SEA) mechanism is proposed to enhance the dim target and suppress the stationary background. Furthermore, a long-term trajectory continuity (LTC) measurement is designed to confirm the real target and eliminate random noises. They fully excavate the temporal features of the slow-moving target.(3)A low-altitude infrared slow-moving small target detection method, namely STFM, is proposed. Experiments were conducted on six real image sequences—the proposed method can achieve a detection probability of 97% at a false alarm rate of 0.01% and performs better than the state-of-the-art methods.

The rest of this article is structured as follows: Section 2 summarizes the related works in the field of infrared small target detection. Section 3 presents the proposed method in detail. Section 4 introduces the dataset and evaluation metrics adopted in the experiment and then analyzes the results. The limitation of the proposed method is discussed in Section 5. Finally, Section 6 concludes this article.

## 2. Related Work

In this section, the related works of single-frame detection and sequential detection methods for infrared small target are briefly reviewed, respectively.

### 2.1. Single-Frame Detection Methods

Single-frame detection methods only exploit the spatial features of the small target or background to enhance the small target or suppress the background. Generally, conventional single-frame methods usually assume that background is consistent in the local spatial domain, i.e., pixels located in the background highly correlate with their adjacent pixels. Under this assumption, some filter-based methods are proposed, including top-hat filter [17], max-mean/max-median filter [18], and two-dimensional least mean square (TDLMS) filter [12]. These methods are sensitive to strong edges and clutters with high-brightness since these structural backgrounds also disrupt the local consistency as small targets do. By exploring the non-local self-correlation property of background while regarding the small target as an outlier in the infrared image, some methods try to transform the small targets detection task into a convex optimization problem of recovering low-rank and sparse matrices that can be solved by principal component pursuit (PCP) [13,19,20,21]; however, these methods are also ineffective in suppressing salient edges and salt noises. Since deep learning technology can automatically learn multi-dimensional features of targets, many deep-learning-based methods have been proposed [22,23,24]; however, if a target only occupies one or two pixels or appears in a low signal-to-noise ratio image, its features is insufficient to be extracted by a deep convolutional network. Recently, another category of traditional single-frame detection methods based on the contrast mechanism of human visual system (HVS) has been proposed, and their practical performance has attracted considerable attention. Chen et al. [7] firstly designed a local contrast method (LCM) by measuring the difference between the central pixel with its nearby pixels using a nine-cell square kernel. Afterward, many improved methods based on LCM are proposed in succession, including the novel local contrast method (NLCM) [8], multiscale patch-based contrast measure (MPCM) [10], variance difference (VARD) measure [25], and double neighborhood gradient method (DNGM) [26]. These HVS-based methods achieve better performance by changing the manner of local contrast measurement or combining other characteristics of the small target. They can effectively enhance the Gaussian-like small target and suppress regular strong edges; however, they will cause missed detection when detecting the point targets. In addition, due to the lack of temporal features, they have a high false alarm rate in case the background contains vary-sized clutter with high brightness.

### 2.2. Sequential Detection Methods

Conventional sequential detection methods, such as temporal variance filter (TVF) [27], temporal hypothesis testing [28], and temporal profiles (Tps) [29], make specific assumptions about the velocity and form of target motion. They are purely temporal, i.e., just operating on the 1-D continuous signal of each pixel in the temporal domain; however, these methods would misidentify the dynamic background (e.g., moving cloud) as the real target; therefore, many sequential methods have been proposed to simultaneously use the properties of 1-D motion features and 2-D spatial features of moving small targets in recent years. Gao et al. [30] combined MPCM and TVF to construct a new filter named temporal variance and spatial patch contrast filter (TVPCF), achieving better performance. Deng et al. [14] defined a spatial-temporal local contrast filter (STLCF) to detect moving targets in infrared image sequences. Du and Hamdulla [15] designed a novel spatial-temporal local difference measure (STLDM) to distinguish target and background by grayscale difference. Pang et al. [16] presented a novel spatial-temporal saliency (NSTS) method by fusing spatial variance saliency measure with temporal gray saliency measure. A new spatial-temporal vector difference measure (STLVDM) was proposed by Zhang et al. [31]. The above methods merge target temporal cues with HVS-based single-frame detection methods and obtain stable performance; however, these methods easily ignore the small target with low contrast because they mainly utilize the interframe difference to distinguish the moving target and the stationary background in the temporal domain. Moreover, they are sensitive to random noises since they do not identify the trajectory continuity of the moving target in an image sequence. Some methods extend the optimization problem on 2-D image patches in the spatial domain to 3-D spatial-temporal tensors (STT) [32,33,34]. For example, a novel edge and corner awareness-based spatial-temporal tensor (ECS-STT) model [35] was presented to suppress the strong edge and corner. These algorithms are more effective, but adopting the cues in the temporal dimension increases the convergence time of them.

## 3. Methodology

In this section, we elaborate on how the proposed method fully utilizes both spatial and temporal features of the infrared small target. Specifically, STFM first calculates the LGD using a circular kernel in a single frame. Then, the temporal saliency map is obtained by the SEA measure on the short-term sub-sequence. Next, The STSM is obtained by fusing the results of the LGD and SEA modules, and the candidate targets are segmented from it through an adaptive threshold mechanism. Finally, the detection result is confirmed by the LTC measure on the long-term sub-sequence. The overall processes are depicted in Figure 1.

### 3.1. Local Grayscale Difference (LGD) Measure

According to the properties of HVS, the primary basis for human recognition of a small target from an infrared image is that the target has apparent discontinuity with its surrounding background [11]. The grayscale distribution reflects the discontinuity of targets in infrared images. To be general, the small target is regarded as a homogenous and compact area of isotropic distribution. Figure 2 shows four typical infrared images captured in various low-altitude scenes and the corresponding 3-D grayscale distributions of the local areas containing the target.

We can see that the target is brighter than its nearby background, and its intensity is locally maximal; therefore, the grayscale difference between the target and its nearby pixels is valuable for detecting the small target. Due to the change in imaging distance and operation environment, the target size is unfixed in practical application. Using the reference to the definition of infrared small target by Society of Photo-Optical Instrumentation Engineers (SPIE) [36], we assume that the size of infrared small targets in low-altitude scenes ranges from 1×1 to 7×7 pixels.

Although square kernels [7,25] are the most commonly used structure to calculate the local contrast in detecting infrared small targets, it cannot fit local symmetric orientations of irregular edges. Inspired by the fact that the retinal ganglion cells in the HVS have roughly concentric receptive fields [37,38], we propose a new circular kernel to measure the local discontinuity of small targets. The circular kernel consists a central pixel and some surrounding pixels that are crossed by a circle with radius *r*. In order to ensure that the surrounding points of the circular kernel can completely cover the background around the target, the relationship between the radius *r* of the kernel and the maximum size of the target need to meet the following requirement:(1)$$r=\sqrt{{\left(\frac{L_{max}}{2}\right)}^{2}+{\left(\frac{3}{2}\right)}^{2}},$$
where $L_{max}$ is the maximum size of small target to be detected. Here, we let $L_{max}=7$ as mentioned above. The circular kernel when $L_{max}=7$ is shown in Figure 3. It consists the central pixel *T* and its surrounding pixels $\{B_{\lambda },\lambda =0,1,2,...,19$}, as shown in Figure 3a. The coordinate $\left\{p_{\lambda },q_{\lambda }\right\}$ of each surrounding point in the circular kernel is shown in Figure 3b. The circular kernel is roughly isotropic, which means it can sample uniformly in different directions over the local region of the original image.

Making full use of the locally maximal and isotropic properties of small target, we propose an LGD measure based on the circular kernel. Given a pixel point (i,j) in the original image, the max gray value of its neighboring background covered by the circular kernel is calculated as follows:(2)$$I_{max}\left(i,j\right)=max\left\{g\left(i+p_{\lambda },j+q_{\lambda }\right),\lambda \in \left\{0,1,2,...,19\right\}\right\},$$
where g(i,j) denotes the gray value at the pixel point (i,j) in the original infrared image. Then, the LGD measure is expressed by Equation (Equation 3), and H(x) represents a step function, which is defined as Equation (Equation 4).
(3)$$I_{\mathrm{LGD}}\left(i,j\right)={\left[g\left(i,j\right)-I_{max}\left(i,j\right)\right]}^{2}\cdot H\left[g\left(i,j\right)-I_{max}\left(i,j\right)\right],$$
(4)$$H\left(x\right)=\left\{\begin{array}{cc} 1 & ,x>0\\ 0 & ,x\leq 0 \end{array}\right.$$

LGD can quantify the discontinuity between the target and its surrounding pixels. Compared with the small target, the grayscale distribution of irregular edges is consistent in a certain direction. In addition, the homogeneous background is consistent with its neighborhood. Thus, the LGD measure can effectively suppress edges and the homogeneous background, as shown in Figure 4. The quadratic operation is a simple way to suppress some residual background [39]; however, due to the technology limitation, infrared sensors may have blind pixels, which causes IR images to contain pixel-sized noises with high brightness (PNHB). In addition, the low-altitude background has more intricate background interference similar to small targets, and their grayscale distribution is isotropic too. It may be hard for people to identify the real target from them just by utilizing information within a single image. So, only using the information in the spatial domain cannot obtain reliable performance; next, we explore temporal clues to distinguish the target from them.

### 3.2. Temporal Features Analysis and Measure

#### 3.2.1. Feature Detection and Image Registration

In practical applications, the infrared detector may shift following the moving target to ensure the target in its FOV, which results in the adjacent frames not being registered. That will make an impact on estimating the real status of the moving target. With consideration of the above situation, interframe registration is employed first. Feature detection and matching is commonly used in image registration. Block matching performs well in [40], but it is inefficient. Corners are important features in an image and useful for patch matching; therefore, the Harris corner detection algorithm [41] is applied in this article, which screens pixels by comparing the eigenvalues of the gradient matrix. The registration process is described as follows: first, two frames are extracted from an image sequence, named base frame $X_{b}$ and test frame $X_{t}$, respectively; then, Harris corners in $X_{b},X_{t}$ are detected; next, the corners are matched [42] and the homography matrix is calculated referring to the well-matched corners; finally, the test frame is registered to the base frame using the homography matrix. The test frame after registration is calculated by
(5)$$X_{t}^{{}^{\prime }}=f_{warp}\left(X_{t},M_{tb}\right),$$
where $f_{warp}$ denotes the geometric transformation, $M_{tb}$ denotes the homography matrix of the test frame $X_{t}$ relative to the base frame $X_{b}$.

#### 3.2.2. Temporal Features Analysis

We further analyze the infrared image sequences and sum up two temporal features of slow-moving small targets: (1) local inconsistency—the appearance of moving targets will cause a local variation of the grayscale in the temporal dimension, and the grayscale of target is the local maxima of one pixel’s temporal profile as shown in Figure 5b; (2) global trajectory continuity—the motion of the slow-moving small target is continuous, i.e., the position of the target will not abruptly change in a video sequence, and the target will appear in the FOV of detector for a long period; in contrast, random clutters are inconsistency.

The improved frame difference (IFD) method has been widely used to obtain TSM in [15,16], which describes that the intensity curve of the target presents a large wave in the temporal domain. So, it employs a simple mechanism. First, select two reference frames from nearby frames of the current frame. Then, the square difference between the maximum intensity and the minimum intensity in the time dimension is utilized as the temporal saliency of the target. The process can be expressed as Equation (Equation 6):(6)$$\left\{\begin{array}{cc} I_{max} & =max\{I_{k-s},I_{k},I_{k+s}\}\\ I_{min} & =min\{I_{k-s},I_{k},I_{k+s}\}\\ I_{\mathrm{IFD}} & ={\left(I_{max}-I_{min}\right)}^{2} \end{array}\right.,$$
where $I_{k-s},I_{k},I_{k+s}$ correspond to the (k−s)th frame, *k*th frame, and (k+s)th frame in an image sequence, respectively, and $I_{\mathrm{IFD}}$ denotes the TSM calculated by IFD method; however, since the IFD method does not fully identify the temporal features of the target, there are three defects: (1) when the target is dim, the difference is slight between $I_{max}$ and $I_{min}$, so it may cause missed detection; (2) it ignores that the grayscale of target is the local maxima, so it will cause a ghost phenomenon, as shown in Figure 5c; (3) it is non-robust because the global trajectory continuity feature is not considered.

#### 3.2.3. Short-Term Energy Aggregation (SEA) Mechanism

Utilizing the local inconsistency feature of a moving small target, we design a SEA mechanism to enhance the dim target by aggregating the grayscale difference between the current frame and its nearby frames. In addition, a truncation measure is adopted to avoid the ghost phenomenon. First, we select (2s+1) frames containing the current frame as a short-term sub-sequence and choose the current frame $I_{k}$ as base frame, other reference frames $\left\{I_{n},n\in k-s,\cdots ,k-1,k+1,\cdots ,k+s\right\}$ need to be registered with $I_{k}$; then, the difference maps between the reference frames and the base frame are calculated, respectively, and the values less than 0 are truncated. Finally, the difference maps are aggregated to obtain a SEA map. The overall processes are summarized as the following equation:(7)$$I_{\mathrm{SEA}}=\sum_{n=k-s}^{k-1}max\left\{I_{k}-f_{warp}\left(I_{n},M_{kn}\right),0\right\}+\sum_{n=k+1}^{k+s}max\left\{I_{k}-f_{warp}\left(I_{n},M_{kn}\right),0\right\},$$
where $I_{\mathrm{SEA}}$ denotes the SEA map, the function $f_{warp}$ has been introduced in Equation (Equation 5), $M_{kn}$ denotes the homography matrix between $I_{k}$ and $I_{n}$.

As shown in Figure 5d, compared with the detection result of IFD method, the SEA mechanism can eliminate the ghost phenomenon while accumulating more energy from several different maps to enhance the dim moving target in the current frame. Moreover, the stationary background interference, e.g., PNHB, can be directly suppressed by image subtraction.

#### 3.2.4. Extract Candidate Targets from STSM

Observing the experimental results of LGD and SEA shown in Figure 4 and Figure 5d, we notice that the intensity in $I_{\mathrm{LGD}}$ and $I_{\mathrm{SEA}}$ both increase in the areas with the real target while weakening in other areas where PNHB and clutters locate. Thus, a normalized fusion mechanism is applied to obtain the STSM. The fusion mechanism can also suppress the boundary effect caused by image registration. It is defined as follows:(8)$$I_{\mathrm{STSM}}=\frac{I_{\mathrm{LGD}}}{max_{i,j}\left\{I_{\mathrm{LGD}}\right\}}⊗\frac{I_{\mathrm{SEA}}}{max_{i,j}\left\{I_{\mathrm{SEA}}\left(i,j\right)\right\}},$$
where ⊗ denotes the pixel-wise multiplication, $I_{\mathrm{LGD}}$ and $I_{\mathrm{SEA}}$ denote the results of Equations (3) and (7), respectively. Then, an adaptive threshold mechanism is applied to segment the candidate targets from STSM. The threshold can be calculated by:(9)$$Th=\xi \cdot max(I_{\mathrm{STSM}}),$$
where max(·) is used to extract the max value of STSM, and ξ is a scale factor ranging from 0.5 to 1 determined experimentally.

#### 3.2.5. Long-Term Trajectory Continuity (LTC) Measure

In practice, sometimes, some clutters similar to the target on the morphology appear randomly, and they vary frequently. Moreover, the image registration may produce a slight error when just a few corners are detected in an image. These are unfavorable to obtaining accurate detection results. Although the STSM can help us locate the target well, it still needs to excavate more information to confirm the final detection results. Inspired by pipeline filter [5] and graph matching [43], we design an LTC measure to eliminate random clutters and registration error by utilizing the global trajectory continuity feature further.

As shown in Figure 6, we use a first-in-first-out (FIFO) queue of length *l* to store a long-term sub-sequence consisting of the current frame and many historical frames. Each candidate target segmented from frames in FIFO is assigned a unique identification (ID) number, e.g., $D_{1},D_{2},D_{3}$ in the current frame. Since the position of the slow-moving target will not change suddenly, we choose the Euclidean distance between the centers of two candidate targets in adjacent frames as the association criterion here. The distance can be calculated as follows:(10)$$d_{\left\{{\mathrm{ID}}_{1},{\mathrm{ID}}_{2}\right\}}=\sqrt{{\left(x_{{\mathrm{ID}}_{1}}^{k}-x_{{\mathrm{ID}}_{2}}^{k-1}\right)}^{2}+{\left(y_{{\mathrm{ID}}_{1}}^{k}-y_{{\mathrm{ID}}_{2}}^{k-1}\right)}^{2}},$$
where $\left(x_{{\mathrm{ID}}_{1}}^{k},y_{{\mathrm{ID}}_{1}}^{k}\right)$ and $\left(x_{{\mathrm{ID}}_{2}}^{k-1},y_{{\mathrm{ID}}_{2}}^{k-1}\right)$ denote the candidate target ${\mathrm{ID}}_{1},{\mathrm{ID}}_{2}$ in the *k*th and (k−1)th frame in an image sequence, respectively.

Take Figure 6 as a specific example to describe the LTC mechanism in detail. When the distance between two targets in adjacent frames is less than a specified distance threshold *R*, we confirm that they are matched successfully and belong to the same trajectory, e.g., the trajectory $\left\{A_{2}\to B_{1}\to C_{1}\to D_{2}\right\}$. Thus, $D_{2}$ in current frame is considered as a true target. Moreover, due to the fact that $C_{3}$ cannot match any candidates in the following frames, it is viewed as random clutter and will be discarded. Further, to ensure the continuity of a track whose length is more than 3, we use the two latest targets in the longer trajectory to predict the position of a missed target in the next frame as follows:(11)$$\left\{\begin{array}{c} {\hat{x}}_{t}^{k}-x_{t}^{k-1}=x_{t}^{k-1}-x_{t}^{k-2}\\ {\hat{x}}_{t}^{k}-y_{t}^{k-1}=y_{t}^{k-1}-y_{t}^{k-2} \end{array}\Rightarrow \left\{\begin{array}{c} {\hat{x}}_{t}^{k}=2x_{t}^{k-1}-x_{t}^{k-2}\\ {\hat{y}}_{t}^{k}=2y_{t}^{k-1}-y_{t}^{k-2} \end{array}\right.\right.$$
where $\left({\hat{x}}_{t}^{k},{\hat{y}}_{t}^{k}\right)$ is the center of the predicted target of *t*th trajectory in the *k*th frame, $\left(x_{t}^{k-1},y_{t}^{k-1}\right)$, and $\left(x_{t}^{k-2},y_{t}^{k-2}\right)$ are the centers of *t*th trajectory in (k−1)th frame and (k−2)th frame, respectively. For instance, the position of $C_{2}$ in Figure 6 is calculated by the positions of $A_{1}$ and $B_{2}$ in the trajectory $\left\{T_{1}\to \cdots \to A_{1}\to B_{2}\right\}$ whose length is more than three. Moreover, since $C_{2}$ matches $D_{1}$ successfully, we confirm the predicted target $C_{2}$ and the candidate target $D_{1}$ are real targets. Finally, there remains a candidate target $D_{3}$ in the current frame that is not matched existing trajectories, we encode it as a start of a new candidate trajectory. Updating FIFO queue, if $D_{3}$ can match another candidate target in the next frame, we confirm it as a true target, such as $A_{2}$; otherwise, it will be discarded, such as $C_{3}$. The details can be found in Algorithm 1.
**Algorithm 1** Long-term Trajectory Continuity (LTC) Measure**Input:**${\left\{D_{j}\right\}}_{N}$—candidate targets extracted from the current frame,   ${\left\{T_{i}\right\}}_{M}$—trajectories of the historical frames stored in FIFO queue,    *R*—distance threshold.**Output:**${\left\{D_{j}^{{}^{\prime }}\right\}}_{N^{{}^{\prime }}}$—confirmed targets in the current frame,    ${\left\{T_{i}^{{}^{\prime }}\right\}}_{M^{{}^{\prime }}}$—updated trajectories in FIFO queue.  1:Initialize two boolean vectors $V_{T},V_{C}$ to store the matching states of each historical trajectory and each candidate target, respectively;  2:**for**i=1 to *M*
**do**  3: **if**
$V_{T}\left(i\right)==False$
**then**  4:  Get the latest target $t_{i}$ in $T_{i}$  5:  **for**
j=1 to *N*
**do**  6:   **if**
$V_{C}==False$
**then**  7:    Calculate the Euclidean distance $d_{i,j}$ between $D_{j}$ and $t_{i}$ by Equation (Equation 10);  8:    **if**
$d_{i,j}<R$
**then**  9:     ${\left\{T_{i}^{{}^{\prime }}\right\}}_{M^{{}^{\prime }}}\leftarrow \left[T_{i},D_{j}\right]$;10:     $D_{j}^{{}^{\prime }}\leftarrow D_{j}$;11:     $V_{T}\left(i\right)\leftarrow True$, $V_{C}\left(j\right)\leftarrow True$;12:    **end if**13:   **end if**14:  **end if**15: **end if**16:**end if**17:**for**i=1 to *M*
**do**18: **if**
$V_{T}\left(i\right)==False$ and $length(T_{i})>3$
**then**19:  Get the position of predicted target $D_{p}$ by Equation (Equation 11);20:  ${\left\{T_{i}^{{}^{\prime }}\right\}}_{M^{{}^{\prime }}}\leftarrow \left[T_{i},D_{p}\right]$;21: **end if**22:**end if**23:**for**j=1 to *N*
**do**24: **if**
$V_{C}\left(j\right)==False$
**then**25:  ${\left\{T_{s}^{{}^{\prime }}\right\}}_{M^{{}^{\prime }}}\leftarrow \left[D_{j}\right]$.26: **end if**27:**end for**

## 4. Experiments and Analysis

In this section, we first introduce the dataset and evaluation metrics used in this article. Then, extensive experiments, including qualitative and quantitative experiments, were conducted to demonstrate the performance of our method. Finally, the ablation study for each module of the proposed method was designed to analyze their effectiveness. We conducted all experiments on a computer with a 2.80-GHz Intel i7-9700 CPU processor and 16.0-GB RAM; the code was implemented in MATLAB 2018a.

### 4.1. Experimental Setup

#### 4.1.1. Datasets

We evaluated the performance of our proposed methods on a public dataset [44] collected by the ATR laboratory of National University of Defense Technology (NUDT). All image sequences contained in the dataset were captured using a mid-wave infrared (MWIR) camera with a resolution of 256×256, and the target is a fuel-powered UAV. The dataset covers low-altitude slow-moving small targets in multiple scenes. Here, we selected six typical image sequences with different scenes to evaluate the effectiveness and robustness of our proposed method. The details of each sequence are described in Table 1.

#### 4.1.2. Evaluation Metrics

Generally, the detection performance is assessed from two aspects, i.e., detection ability and detection precision. The SCR gain (SCRG) and background suppression factor (BSF) evaluate the detection ability by measuring the difference between the saliency map and the original image. They are the most commonly used metrics in the field of infrared small target detection. The SCRG can measure the degree of target enhancement, defined as follows:(12)$$\mathrm{SCRG}=\frac{{\mathrm{SCR}}_{sal}}{{\mathrm{SCR}}_{ori}}$$
where ${\mathrm{SCR}}_{sal}$, ${\mathrm{SCR}}_{ori}$ denote the SCR of the saliency map and the original image, respectively. Moreover, the SCR is defined as follows [13]:(13)$$\mathrm{SCR}=\frac{\left|m_{t}-m_{b}\right|}{\sigma_{b}}$$
where $m_{t}$ and $m_{b}$ denote the average gray of the target area and its nearby background area, and $\sigma_{b}$ represents the grayscale standard deviation of the nearby background area. SCR also can describe the difficulty of detection. In general, the lower the SCR of a small target is, the harder it can be to detect. Assume that the size of a small target is a×b, then the nearby background area refers to a hollow local region with the width of *d* as shown in Figure 7. Here, we set d=10.

The BSF measures the background suppression ability by comparing the discrete degree of the background grayscale distribution in the saliency map and original image, defined as follows:(14)$$\mathrm{BSF}=\frac{\sigma_{ori}}{\sigma_{sal}}$$
where $\sigma_{ori}$, $\sigma_{sal}$ are the gray standard deviation of the background in the original image and in the saliency map. The higher the BSF of a method is, the better it can suppress noise and clutters; however, when the background in the saliency map is very clean, the denominator in Equation (Equation 13) or Equation (Equation 14) may equal zero. Then, the values of SCRG and BSF are infinity (Inf) and unable to quantify the detection ability of methods. To avoid this case, we also adopt local contrast gain (LCG) [45] to evaluate the detection ability of each method. The LCG is defined as follows:(15)$$\mathrm{LCG}=\frac{{\mathrm{LC}}_{sal}}{{\mathrm{LC}}_{ori}}$$
where ${\mathrm{LC}}_{sal}$ and ${\mathrm{LC}}_{ori}$ denote the local contrast (LC) of the target in the saliency map and original image, respectively. In addition, the LC is calculated as follows:(16)$$\mathrm{LC}=|m_{t}-m_{b}|$$
where $m_{t}$ and $m_{b}$ are the same as those in Equation (Equation 13).

Moreover, we employ the receiver operating characteristic (ROC) curve to evaluate the detection precision of methods. ROC curve describes the relationship between the detection probability ($P_{d}$) and false alarm rate ($F_{a}$), which are defined as follows:(17)$$P_{d}=\frac{N_{\mathrm{DT}}}{N_{\mathrm{AT}}}$$
(18)$$F_{a}=\frac{N_{\mathrm{FA}}}{N_{p}}$$
where $N_{\mathrm{DT}}$ denotes the number of detected true targets, $N_{\mathrm{AT}}$ denotes the total number of actual targets in an image sequence, $N_{\mathrm{FA}}$ denotes the number of detected false alarm pixels, and $N_{p}$ denotes the total number of pixels in an image sequence. Most of the targets in the dataset are point targets, and the labels provided only contain the coordinate of the target center without the height and width of the targets; therefore, refer to [44], if the detected target contains a labeled center and the distance between the center of detected target and the labeled center within three pixels, the detected target can be regarded as a true target. Otherwise, it will be regarded as a false alarm target.

#### 4.1.3. Baseline Methods

In order to demonstrate the practical and robust performance of the proposed method, some classical single-frame detection methods and existing sequential methods were chosen as the compared baseline methods. The single-frame detection methods include new white top-hat (NWTH) filter [46], MPCM [10], and RIPT [19]. The sequential detection methods includes STLDM [15], NSTS [16], and STLVDM [31]. The NWTH filter proposed a solid circular structure and a hollow circular structure, respectively, and combining the morphological operation to suppress complex background. MPCM is a popular HVS-based method using a multiscale nine-cell square kernel. RIPT is an improved IPI-based method that introduces the local structure prior knowledge to suppressing edges and enhancing the dim target. STLDM, NSTS, and STLVDM are the excellent sequential detection methods modifying the IFD method described in Equation (Equation 6) and combining local contrast measurement. The parameter settings of these methods are listed in Table 2, and all parameters have been adjusted to obtain the best results.

### 4.2. Qualitative Comparison

Figure 8, Figure 9 and Figure 10 show the representative frames in Seq.1–Seq.6, and the corresponding saliency maps processed by different methods. The real and detected target regions are marked with red boxes; in contrast, the missed targets are marked with yellow boxes.

The NWTH filter can suppress flat background, but it is sensitive to clutters resembling the small target in size and brightness. The MPCM can effectively detect the Gaussian-like targets, as shown in Figure 8; however, MPCM adopted a nine-cell square structure, so it cannot sample uniformly in different directions. There are some residual edge clutters in its saliency map when the background, such as Seq.4 and Seq.5, contains irregular edges. Moreover, it is easy to ignore the pixel-sized target due to the mean operation. The RIPT introduced edge features as prior information to suppress edges and preserve the dim target, so it suppresses edges better than MPCM; however, the results processed by RIPT shown in Figure 8 are hollow due to the uniform grayscale distribution inside the target. The RIPT is also sensitive to target-like clutters, and the residual clutters are even more prominent than the actual target in its saliency maps, which will cause a high false alarm rate. Compared with single-frame detection methods, STLDM and NSTS obtain relatively better performance. The enhancement effects of STLDM on targets with low SCR are unsatisfactory because it only considers the local gray differences in the spatial-temporal domain but does not accumulate the energy of targets in successive frames. Although STLVDM is a sequential detection method, it is ineffective under the slowly changing background and has noticeable ghost phenomenons. In contrast, the proposed method can not only detect the target with tiny size and low SCR, but also suppress various background clutters and obtain stable performance against the slowly changing background.

### 4.3. Quantitative Comparison

To further demonstrate the detection performance of the proposed method, we use SCRG, BSF, and LCG to quantitatively analyze the results of different methods on Seq.1–6. The detailed results are listed in Table 3, and the $\bar{\mathrm{SCRG}}$, $\bar{\mathrm{BSF}}$, and $\bar{\mathrm{LCG}}$ represent the averages of the abovementioned metrics on an image sequence, respectively. We can observe that the $\bar{\mathrm{SCRG}}$ and $\bar{\mathrm{BSF}}$ values of the proposed method on Seq.1, Seq.3, and Seq.4 are Inf, which means that the background clutters near the actual target are completely suppressed. The NWTH filter obtains the saliency map by subtracting the predicted background map from the original image, so its $\bar{\mathrm{LCG}}$ values are less than 1 when the target has high brightness. The MPCM has a relatively weak ability to suppress various low-altitude background clutters since its $\bar{\mathrm{BSF}}$ values are the lowest. Moreover, the $\bar{\mathrm{LCG}}$ values of the RIPT on Seq.1 and Seq.2 are smaller mainly due to the hollow effect, as shown in Figure 8. In contrast, the sequential detection methods simultaneously use the information in the spatial and temporal domains, thus achieving better results. Although STLVDM achieves the highest $\bar{\mathrm{LCG}}$ on Seq.1, Seq.2, and Seq.4, it has poor background suppress performance. The reason is that there are many residual background clutters in its saliency maps because of the ghost phenomenon. Compared with the baseline methods, the proposed method has the largest values of $\bar{\mathrm{SCRG}}$ and $\bar{\mathrm{BSF}}$ on all sequences since it can almost suppress the background clutters completely. In general, the quantitative experimental results demonstrate the proposed method has better target enhancement ability and background suppression ability.

Furthermore, the ROC curves on Seq.1–6 are given in Figure 11 to demonstrate the advantage of the proposed method in detection precision. ROC curve is a common means for visualizing the detection performance. In addition, the curve near the upper-left means the method can maintain high $P_{d}$ with low $F_{a}$. It is obvious that the proposed method achieves better detection precision than other baseline methods on different image sequences; therefore, it is reasonable to conclude that the proposed method has robust performance in various scenes. Moreover, the ROC curves of MPCM on Seq.3–6 are close to the horizontal axis since the MPCM is ineffective in detecting the point target.

### 4.4. Ablation Study

The proposed method contains three modules: LGD, SEA, and LTC. Here, we design ablation experiments to analyze the contribution and practicality of each module to the detection performance. We test different combinations of modules in the proposed method on all test image sequences containing Seq.1–6. The ROC curves are shown in Figure 12, and the $\bar{P_{d}}$, $\bar{F_{a}}$ denote the average detection probability and the average false alarm rate of all test image sequences, respectively. Moreover, we also compare their performance via the $\bar{P_{d}}$ with a constant false alarm rate [47], and the details are listed in Table 4. Note that all experiments were conducted after image registration described in Section 3.2.1. We can find that each module of the proposed method contributes to promoting detection performance, and all modules together bring the optimal result. The LGD measure using the circular kernel shown in Figure 3 preserves the point target as much as possible in the spatial domain. The SEA module effectively enhances dim targets by accumulating multiple interframe differences in the temporal domain. Significantly, the LTC mechanism can further eliminate many random false alarms based on the trajectory continuity of the slow-moving target; therefore, the proposed method obtains a higher $\bar{P_{d}}$ with lower $\bar{F_{a}}$. This also demonstrates that our method fully excavates the features of the low-altitude slow-moving small target both in spatial and temporal domains, and achieves a more satisfactory detection precision.

## 5. Discussion

The qualitative and quantitative experiments both demonstrate the advantage of the proposed method in detecting the low-altitude infrared moving small target. However, it still has limitations in detecting the target submerged by high-brightness background clutter for a long period or the target under blurry background. Figure 13 gives two examples in which the target is submerged by high-brightness background clutter, and the background is blurry due to the fast movement of the detector, respectively. Since the grayscale difference between the target and its surrounding background is tiny and even equal to zero, as shown in Figure 13a, it is also difficult for human to distinguish the position of the actual target area. In this case, the LGD measurement is invalid and the target is missed in $I_{\mathrm{LGD}}$. Moreover, since the target is submerged for a long period, the association mechanism in LTC measurement is ineffective; thus, our method fails to detect it. In another case, as shown in Figure 13b, the rapid movement of the target or the detector FOV will blur the images. This causes the size of the target to exceed the definition of an infrared small target, leading to the missed detection in the $I_{\mathrm{LGD}}$ map. Moreover, there are not enough feature points in the blurry background to complete the image registration between the adjacent frames, which results in numerous residual background clutters in the $I_{\mathrm{SEA}}$ map. In future work, we explore an effective position prediction mechanism to track the target submerged in background for a long period and introduce a practical image deblurring algorithm to enhance image registration performance.

## 6. Conclusions

In this article, we propose an effective low-altitude infrared slow-moving small target detection method, namely STFM. The main idea of it is to promote the target enhancement ability and background suppression ability by integrating the results obtained in the spatial and temporal domains, and improve the detection precision through multi-frame confirmation; therefore, we first construct a circular kernel to calculate the LGD in a single frame. Then, a SEA mechanism is proposed to accumulate the energy of the moving target in several successive frames. Finally, we design an LTC module to confirm the real target and eliminate false alarms. Moreover, the interframe registration technique is introduced to eliminate the interference of background motion. Our method fully excavates the spatial-temporal features and trajectory continuity of the slow-moving target. Extensive experiments were conducted on six image sequences containing various real low-altitude scenes. The results demonstrate that our method has satisfactory performance both in detection ability and detection precision, and outperforms the existing excellent single-frame and sequential detection methods.

## Figures and Tables

**Figure 1 sensors-22-05136-f001:**
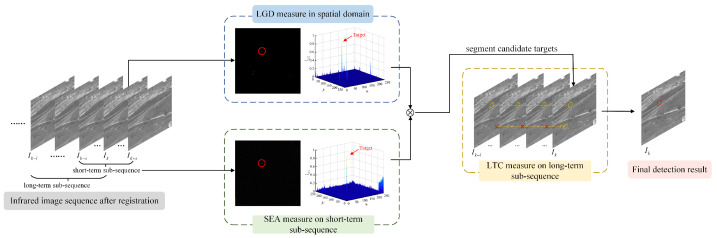
Flow chart of the proposed STFM method. Here, $I_{k}$ denotes the *k*th frame in the image sequence. The short-term sub-sequence contains (2s+1) frames and the long-term sub-sequence contains *l* frames (l>s). The candidate targets and the real target are marked using yellow circles and a red circle, respectively, and the red multiplication sign means that the candidate targets in adjacent frames are matched unsuccessfully.

**Figure 2 sensors-22-05136-f002:**
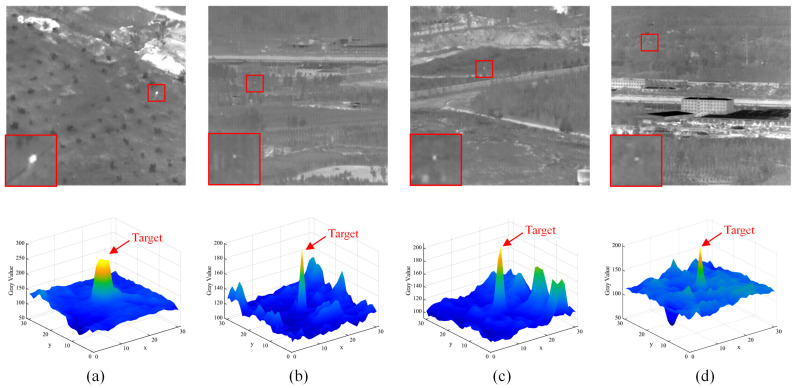
Representative of infrared small targets in various low-altitude scenes. The first row of (**a**–**d**) shows the original images, and the actual target area (less than 7×7 pixels) is marked using a red box with a close-up version shown in the bottom-left of each image. The second row of (**a**–**d**) shows the grayscale distribution (3D-view) of of the local areas containing the target.

**Figure 3 sensors-22-05136-f003:**
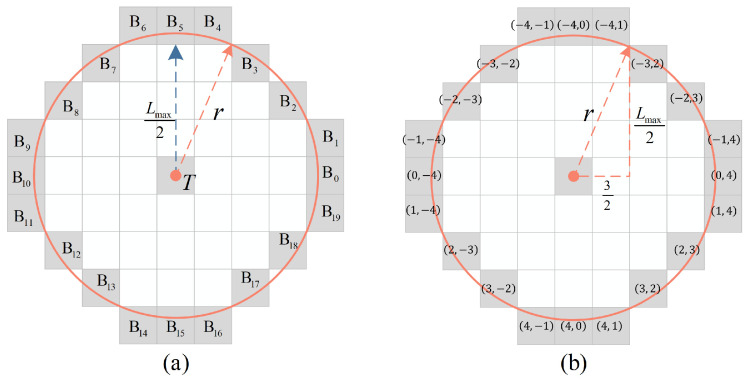
The structure of the circular kernel when $L_{max}=7$. (**a**) The central point *T* and each surrounding point $B_{\lambda }$ in the circular kernel. (**b**) The corresponding coordinate $\left(p_{\lambda },q_{\lambda }\right)$ of each point in (**a**).

**Figure 4 sensors-22-05136-f004:**
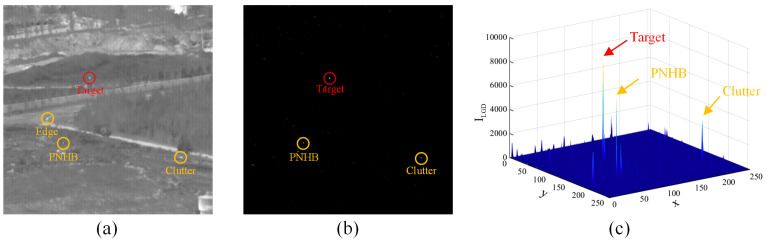
An example of LGD result. (**a**) The original infrared image. (**b**) The saliency map of LGD measure. (**c**) The 3D view of (**b**).

**Figure 5 sensors-22-05136-f005:**
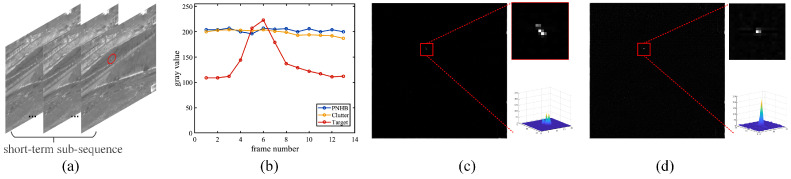
An example of a short-term sub-sequence. (**a**) The short-term sub-sequence containing registered images. (**b**) The grayscale curves of different areas in the local temporal dimension. (**c**,**d**) The detection results of the IFD method and SEA method, and a red box marks the actual target area with a close-up version and its 3D view shown on the right-top and right-bottom, respectively.

**Figure 6 sensors-22-05136-f006:**
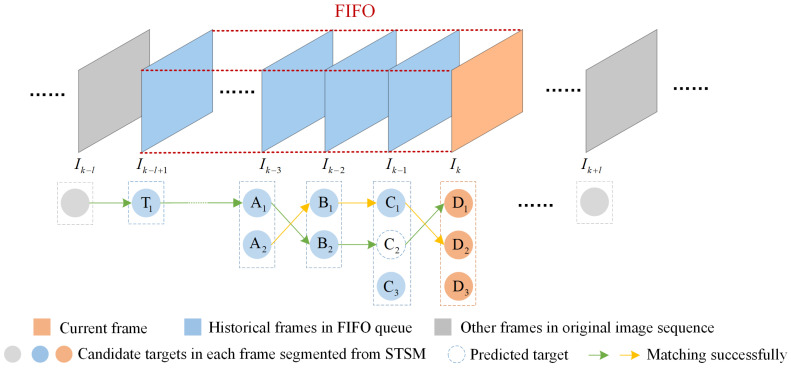
Illustration for LTC measure. Lines of the same color with an arrow mean that candidate targets in adjacent frames are matched successfully.

**Figure 7 sensors-22-05136-f007:**
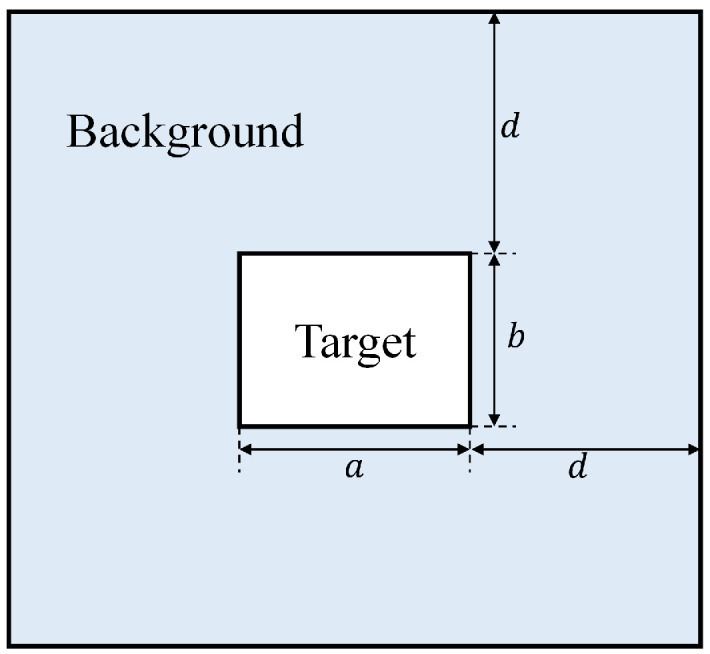
Illustration of target area and its nearby background area.

**Figure 8 sensors-22-05136-f008:**
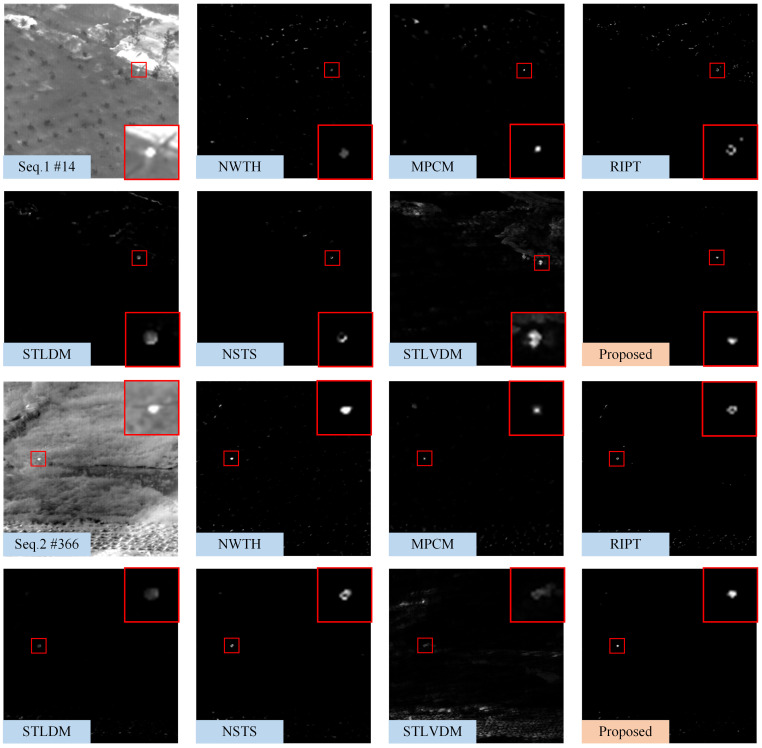
The saliency maps obtained by different methods on 14th frame in Seq.1 and 366th frame in Seq.2. For the sake of clarity, the target regions are marked with red boxes, and a close-up version is shown on each map.

**Figure 9 sensors-22-05136-f009:**
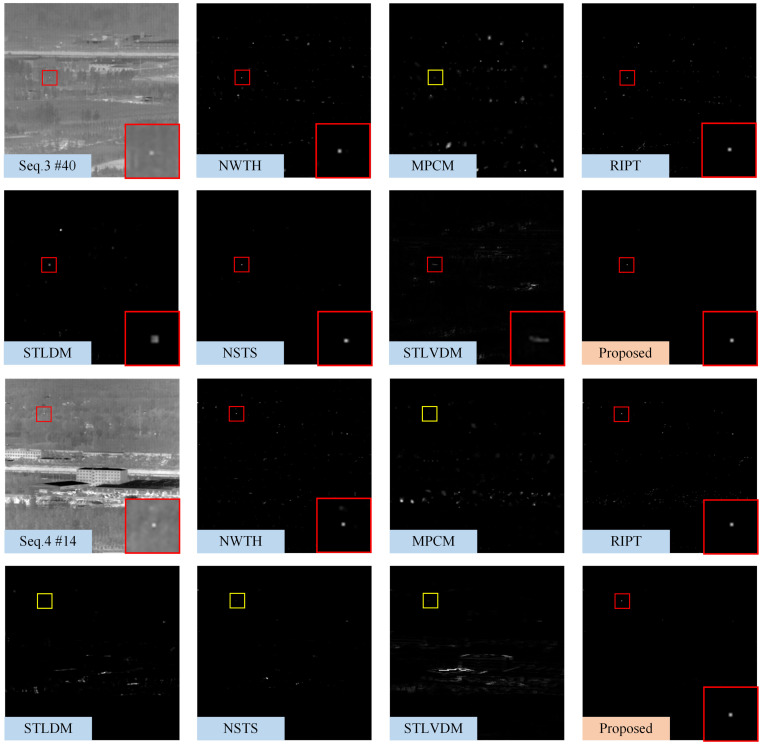
The saliency maps obtained by different methods on 40th frame in Seq.3 and 14th frame in Seq.4. For the sake of clarity, the real and detected target regions are marked with red boxes, and a close-up version is shown on each map. The yellow boxes denote missed detection.

**Figure 10 sensors-22-05136-f010:**
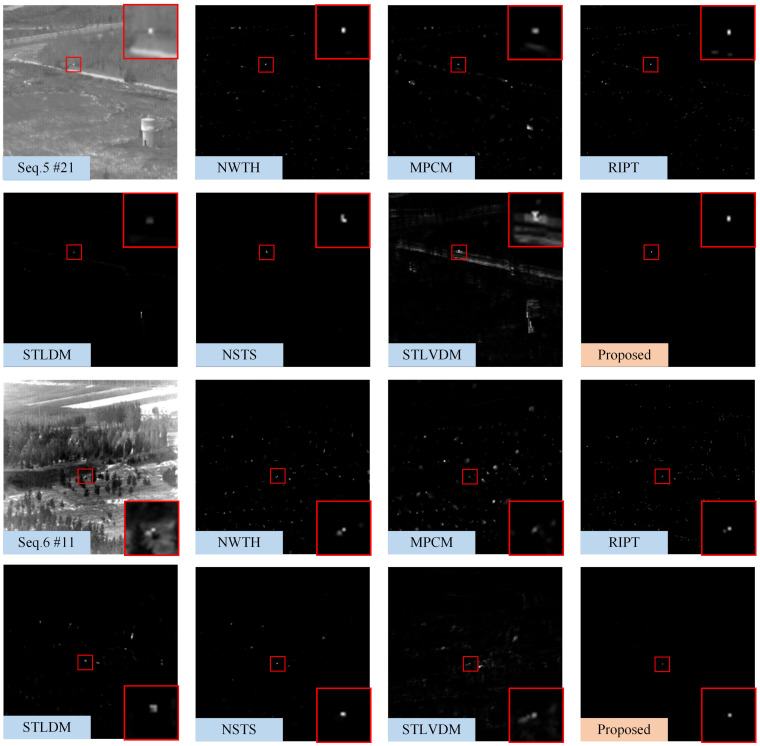
The saliency maps obtained by different methods on 21th frame in Seq.5 and 11th frame in Seq.6. For the sake of clarity, the target regions are marked with red boxes, and a close-up version is shown on each map.

**Figure 11 sensors-22-05136-f011:**
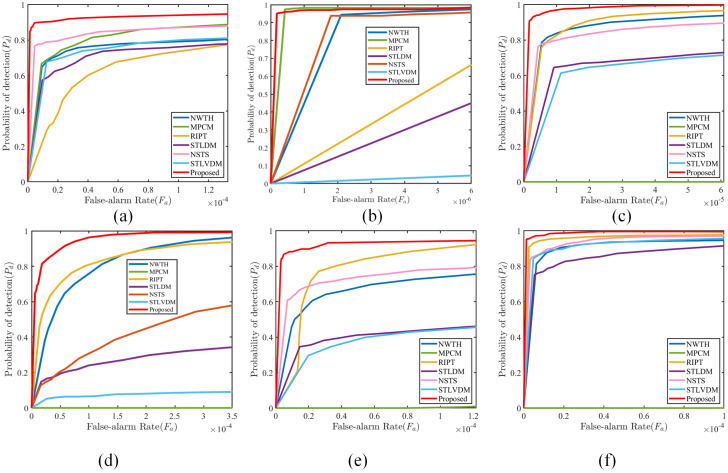
(**a**–**f**) ROC curves of different methods on Seq.1–6, respectively.

**Figure 12 sensors-22-05136-f012:**
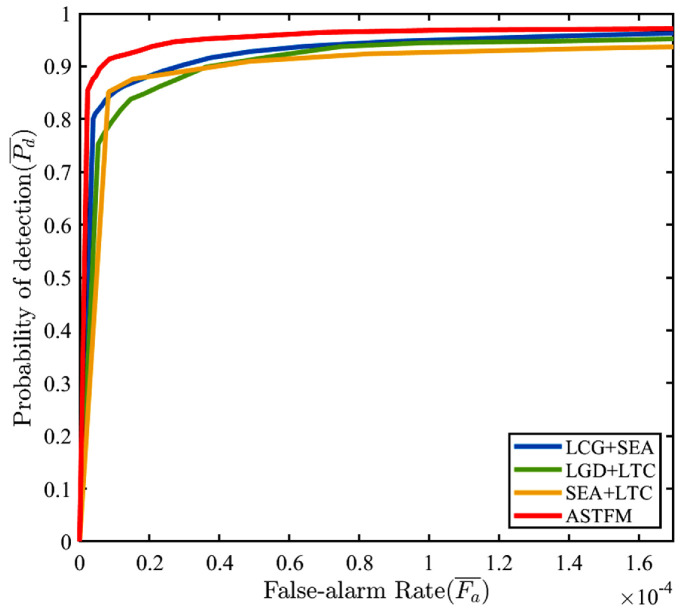
ROC curves of different combinations of modules in the proposed method on all test image sequences containing Seq.1–6.

**Figure 13 sensors-22-05136-f013:**
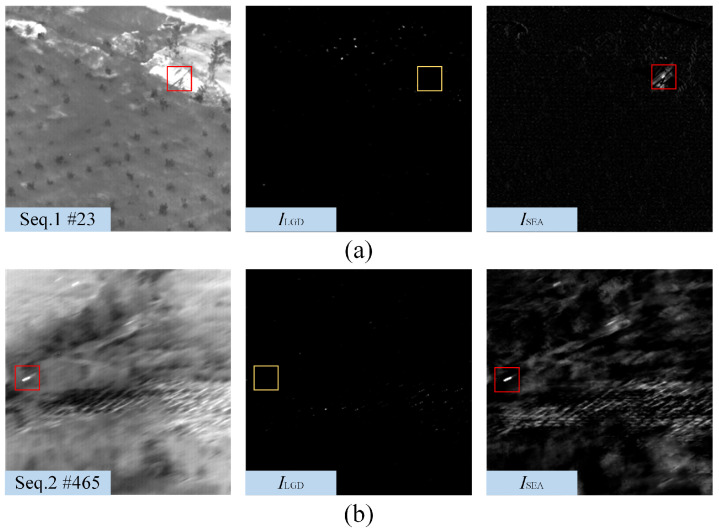
Two examples of missed detection. The original image, $I_{\mathrm{LGD}}$ map, and $I_{\mathrm{SEA}}$ map are shown in turn, and the actual target area is marked with a red box, the missed target is marked with a yellow box. (**a**) The target submerged by the high-brightness background. (**b**) The target under blurry background.

**Table 1 sensors-22-05136-t001:** Details of different image sequences.

Seq.	Frame Number	Target Size	Scene Description
1	399	4×4	Single target from near to far, non-uniform background
2	500	3×3∼6×7	Single target from far to near, chaotic ground background
3	1500	1×1∼2×2	Single target from far to near, cluttered ground background
4	751	1×1∼2×2	Single target, ground background containing man-made buildings
5	399	3×3∼1×2	Single target from near to far, non-uniform background
6	500	1×2	Single dim target, cluttered ground background

**Table 2 sensors-22-05136-t002:** Parameter settings of baseline methods.

Methods	Parameter Settings
NWTH	$M\left(\Delta B\right)=4,S\left(B_{i}\right)=7$
MPCM	Cell size: 3×3, 5×5, 7×7, 9×9
RIPT	Patch size: 30×30, sliding step: 10, L=0.7, h=1, ϵ=0.01, $\varepsilon =10^{-7}$
STLDM	Subblock size: 3×3, l=5
NSTS	Internal cell size: 3×3, middle cell size: 7×7, external cell size: 11×11, n=5
STLVDM	p=3, l=4
STFM (proposed)	$L_{max}=7$, s=4, l=10, R=7

**Table 3 sensors-22-05136-t003:** Quantitative comparison of different methods on Seq.1–6.

Seq.	Metrics	NWTH	MPCM	RIPT	STLDM	NSTS	STLVDM	Proposed
1	$\bar{\mathrm{SCRG}}$	21.34	20.45	6.98	40.84	282.74	6.30	**Inf**
	$\bar{\mathrm{BSF}}$	20.11	1.42	35.90	25.39	90.05	4.75	**Inf**
	$\bar{\mathrm{LCG}}$	0.86	0.82	0.18	1.64	0.81	**1.79**	1.48
2	$\bar{\mathrm{SCRG}}$	17.5486	12.01	4.22	32.20	182.49	4.22	**350.63**
	$\bar{\mathrm{BSF}}$	34.40	1.29	26.66	21.36	Inf	4.10	**401.66**
	$\bar{\mathrm{LCG}}$	0.54	0.87	0.17	1.24	1.00	**1.57**	1.31
3	$\bar{\mathrm{SCRG}}$	37.02	4.28	17.40	9.19	44.82	19.67	**Inf**
	$\bar{\mathrm{BSF}}$	50.20	1.64	31.18	6.35	58.82	20.97	**Inf**
	$\bar{\mathrm{LCG}}$	1.41	1.53	0.50	3.15	3.06	4.30	**5.64**
4	$\bar{\mathrm{SCRG}}$	23.93	2.38	6.68	4.04	33.58	5.54	**Inf**
	$\bar{\mathrm{BSF}}$	23.75	1.23	10.37	24.77	14.53	10.75	**Inf**
	$\bar{\mathrm{LCG}}$	0.89	0.30	0.54	1.85	1.24	1.13	**1.97**
5	$\bar{\mathrm{SCRG}}$	20.44	3.95	18.15	19.31	31.94	16.97	**325.77**
	$\bar{\mathrm{BSF}}$	29.81	1.46	19.96	9.5	16.92	7.19	**261.93**
	$\bar{\mathrm{LCG}}$	0.66	0.75	0.72	2.37	3.68	**4.04**	2.08
6	$\bar{\mathrm{SCRG}}$	11.87	7.79	26.76	15.51	31.80	17.6547	**278.18**
	$\bar{\mathrm{BSF}}$	10.96	1.12	21.19	4.54	3.52	2.6264	**282.64**
	$\bar{\mathrm{LCG}}$	1.06	0.82	2.11	2.28	2.30	3.22	**3.56**

The maximum value of each line is highlighted in bold.

**Table 4 sensors-22-05136-t004:** Ablation study on different combinations of modules in the proposed method ($\bar{F_{a}}=0.01\%$).

Modules			$\bar{P_{d}}$(%)
LCG	SEA	LTC	
✓	✓		95.65
✓		✓	94.72
	✓	✓	93.52
✓	✓	✓	97.11

## Data Availability

The data used in experiments are available at: https://www.scidb.cn/en/detail?dataSetId=720626420933459968dataSetType=journal, accessed on 3 June 2022.

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
