# Peer review of "Low-Altitude Infrared Slow-Moving Small Target Detection via Spatial-Temporal Features Measure"

_sensors, 2022, doi:10.3390/s22145136_

Round 1

Reviewer 1 Report

The following is the review:

Summary: the manuscript describes improvements using UAV (unamanned autonomous vehicles) to be able to detect targets in imagery. The major breakthrough would be in the application of a spatial-temporal approach that would incorporate both the spatial and temporal variability of the target being observed. My comments (more from a big picture perspective ) are found below.  My overall recommendation is that there is merit in the manuscript, but needs some revision.

Points:

Major: I have to leave the review of the methodologies to others who have more experience. Is there somewhere a definition of the target that is trying to be identified? It is still a bit unclear whether the methodologies can be applied to different “targets”? Can the authors make comments about whether any satellite technology could be used in the implementation of the algorithms themselves. The authors make some good points for future research, is this a possibility? The validation appears to be very qualitative overall. Is it possible to quantify more?

Minor instead of the word excavate please use “idenfity”.

There are a tremendous amount of acronyms used. The manuscript would be much easier to read if a table of all the arconymns was added.

Author Response

Dear Reviewer,

       Thank you very much for your careful check and review. Those comments and suggestions are all valuable and helpful for revising and improving our manuscript. We have carefully studied the comments and made some corresponding revisions to our manuscript. The point-by-point responses to the comments are listed in the response letter. Further, all the changes are noted by blue marks in the revised manuscript. Please see the response letter and the revised manuscript for details.

Yours sincerely,

Jing Mu

E-mail: mujing@mail.sitp.ac.cn

Reviewer 2 Report

This is a very nice article that I have read in a long time. I think the manuscript is good enough to be accepted in its current form. 

There are some typographical errors which can be taken care of during proof-reading and final publication.

Author Response

Dear Reviewer,

We gratefully appreciate your positive comment and recognition of our manuscript entitled “Low-Altitude Infrared Slow-Moving Small Target Detection via Spatial-Temporal Features Measure” (ID- sensors-1780283). We will actively cooperate with the editor to correct the typographical errors before the final publication. Thank you again for reading our paper carefully.

Best regards,

Jing Mu.

Reviewer 3 Report

This paper proposed a low altitude slow-moving small target detection method based on the spatial-temporal features measure. 

1) There are too many acronyms in the article, which makes it difficult to read.

2)In line 187, k is fixed at 19 for the different values of r? How to get B_k for different values of r?

3) In equation 1, what's the value of r when L_max is set to 7? 

4) In line 225, how to match the corner points?

5) In equation 9, what is the value of k in the experiment?

6) how do the different values of s, l, and R affect the results in the experiment?

Author Response

Dear Reviewer,
Thank you very much for your careful check and review. Those comments and suggestions are all valuable and helpful for revising and improving our manuscript. We have carefully studied the comments and made some corresponding revisions to our manuscript. The point-by-point responses to the comments are listed in the response letter. Further, all the changes are noted by blue marks in the revised manuscript. Please read the revised manuscript and the response letter for details.

Yours sincerely,
Jing Mu
E-mail: mujing@mail.sitp.ac.cn

Round 2

Reviewer 3 Report

The revised paper is clearer to read and has been modified according to my suggestions.  I think it can be accepted.